# Implementation of an Embedded System into the Internet of Robotic Things

**DOI:** 10.3390/mi14010113

**Published:** 2022-12-30

**Authors:** Jakub Krejčí, Marek Babiuch, Ján Babjak, Jiří Suder, Rostislav Wierbica

**Affiliations:** 1Department of Robotics, VSB—Technical University of Ostrava, 708 00 Ostrava, Czech Republic; 2Department of Control Systems and Instrumentation, VSB—Technical University of Ostrava, 708 00 Ostrava, Czech Republic

**Keywords:** Internet of Things, Internet of Robotic Things, smart sensors, embedded systems, MQTT, ESP32, microcontroller, Nicla, Thingworx

## Abstract

The article describes the use of embedded systems in the Industrial Internet of Things and its benefits for industrial robots. For this purpose, the article presents a case study, which deals with an embedded system using an advanced microcontroller designed to be placed directly on the robot. The proposed system is being used to collect information about industrial robot parameters that impact its behavior and its long-term condition. The device measures the robot’s surroundings parameters and its vibrations while working. Besides that, it also has an enormous potential to collect other parameters such as air pollution or humidity. The collected data are stored on the cloud platform and processed and analysed. The embedded system proposed in this article is conceived to be small and mobile, as it is a wireless system that can be easily applied to any industrial robot.

## 1. Introduction

Nowadays, the concept of Internet of Robotic Things (referred to as IoRT) is increasingly promoted, and its technological aspects, features and opportunities are presented. In our article, we present a concrete implementation of monitoring robotic data and an analysis system in the Thingworx system and place our application in the context of IoRT with its individually described functionalities. In this section presenting the current state of the described issue, we will analyse the available sources giving an overview of IoRT, including its architectural basis, using hardware and software components, the communication protocols used, cloud services, middleware application platforms, and all related technologies behind which our application deployment is based on the field of IoRT. The term Internet of Things (referred to as IoT) gradually became popular in all technological areas of use [1], in the areas of wearable electronics, health care [2], agriculture [3], smart home automation [4], mobile robot applications [5] and also in other advanced branches of industry. This concept had to be gradually standardised, and the logical outcome was the involvement of the leading network and communication technologies leaders to create their architectural model [6]. In the ever-growing industry, with its advanced automation and the setting up of robotics in all industrial areas of production, IoRT has become increasingly talked about. This concept is mainly the connection of IoT, robotics, Cloud Computing and the use of Edge Computing [7,8]. In more detail, the article [9] defines Cloud computing, Edge Computing and Fog Computing, including the embedding in the architecture. Advanced cloud/fog/IoT computing patterns to maximise manufacturing productivity in industrial factories is described in the article [10], which also proposes a model for evaluating the performance of integrated IoRT systems with fog-cloud computing paradigms. A comparative overview of the existing cloud solutions for robotic applications from the point of view of edge computing and related computational technologies is described in [11]. The IoRT architectural model was gradually expanded by additional layers over time with defined requirements. From the original three-layer model [12], seven essential layers have been settled upon [13] for which the individual standards, networking technologies, key features and protocols are defined. In the already comprehensive IoRT architectural concept, we have several overview analyses: An evaluation from the point of view of existing robotic typologies, robotic equipment, existing processing units and robotic cloud platforms is described by [14]. A very clear analysis of IoRT from the point of view of the manufacturing domain and technology is given by [15]. A comprehensive perspective of IoRT architecture with the precise placement of IoRT platforms in a layered model and a description of IoRT interoperability is described in [16]. This detailed review describes an overview of the IoRT concept, technologies, architectures and applications and provides comprehensive coverage of future challenges, developments and applications, including complex IoT open platform architecture requirements. Last but not least, a helpful IoRT classification according to its physical operation, the origin of robotics technology employed, and application areas with a comprehensive overview of different works involving approaches to IoRT enabling technologies is described in [17].

Our solution relies on the PTC ThingWorx platform. ThingWorx is a platform that allows the rapid creation of complete solutions for an industrial IoT [18]. A comparative analysis of some platforms, including Thingworx, is dealt with in [19]. The IoRT architecture is excellently analysed in terms of data exchange techniques with this comparative analysis, giving the advantages, and disadvantages of different data exchange methodologies in article [20]. We use the Message Queuing Telemetry Transport protocol (referred to as MQTT) communication at the application layer [21]. The study [22] provides a comparison with the Advanced Message Queuing Protocol. The continued growth of more demanding robotic applications in data transmission and analysis brings opportunities and challenges for implementing 5G networks [23]. Article [24] describes various use cases and discusses simulation results of a 5G-based robotics application in an industrial environment. Furthermore, in 5G networks, the increasing involvement and development of artificial intelligence, machine learning and deep learning are also expected, which is clearly analyzed in [25]. Among other things, other topics associated with IoRT are also being developed, for example, in the social science field. The work [26] is devoted to researching how an organisational structure with a high occurrence of IoRT elements can be understood and what consequences it can have on the perception of IoRT. This is a non-negligible area of current research because of the penetration of IoRT into people’s everyday life and the share of robotic applications in households. Therefore, it is necessary to assess the consequences on the quality of life and the influence of people’s perceptions of each other in the workplace and households [27].

This work comes up with the design of an embedded device placed on an industrial robot to collect data from the robot and its surroundings. The thesis starts by describing the process of selecting the components that make up the final device. Subsequently, the parameters of the proposed device are explained. Then, the paper describes the process of transferring and storing the collected data in the cloud. Finally, it describes how the data is further handled and what possibilities and advantages this system brings.

## 2. Literature Review

This section discusses the literature that is relevant to the areas covered in this article. The reference literature is divided into Table 1, Table 2 and Table 3. Each table then gives an overview of the area under study and provides references that deal with the issue in more detail. Table 1 reviews articles that deal with general topics in the IoRT concept. It discusses the communication protocols that are used, the challenges and future trends, as well as applications. Table 2 describes hardware and software related to Industry 4.0 and IoRT. The table describes articles that focus on the comparison of platforms used in IoT and the analysis of the hardware structure. It then describes articles that discuss software tools for program development and practical implementation options. Finally, Table 3 contains articles that summarize comparisons of IoT platforms, some of which use our Thingworx software, MQTT protocol and describe case studies of robotic cloud applications.

The work in this paper is based on our experience in the subject and also on the requirements for the system. We support our ideas and approaches by analysing the current state of the art, which we provide in the tables below.

## 3. Software and Hardware Used

The starting point for the whole system is the industrial robot’s workstation located in the university’s laboratories of the Department of Robotics. The workplace in Figure 1 is designed as a robotic cell, where an ABB IRB 1600 robot is installed. The workstation contains several security features, from which the workstation fencing and laser scanners can be seen at a glance. The industrial robot is placed in the centre of the workstation on a stand, carrying a positioner. The robot is equipped with a force sensor and simulates a deburring operation. This is a repetitive operation where the force sensor is used to monitor the pressure of the effector on the workpiece. The robot’s control units are next to the workstation, which controls the robot.

As a complex mechanism, an industrial robot is influenced by many parameters. This system must be regularly inspected and serviced to prevent undesirable situations. Typically, only trained persons should operate an industrial robot workstation. The control unit directs the robot’s movements in relation to the program to be executed. It also supervises parameters such as the speed, rotation and torque of the individual joints of the industrial robot, which make up the industrial robot. The described industrial robot consists of six joints which are each loaded asymmetrically. Due to inaccuracies, degradation of the system due to time, overloading of the system, and other negative influences such as temperature drift of the robot, oscillations of the robot arm and, thus, potential inaccuracies in the robot’s work performance occur.

In the work described in our article, we use several software tools. For controlling and programming the robot, we use software from the robot manufacturer ABB RobotStudio. In this simulation system, the robot’s working operations are designed, and its parameters are set. For connecting the control unit to the network, we use ABB IRC5 OPC Server software. In this way, all robot systems in the laboratory are connected. Middleware from the Kepware family of PTC systems is used for further data movement. This handles the processing of the monitored data, which is passed on to other software from the PTC family Thingworx. This cloud platform is used as the controller of the entire IoT system; and there will be more on it later in the article.

## 4. IoRT System Application

This chapter will discuss the proposed solution and its use with an industrial robot. The system is designed as a loop which gathers data from two different sources and collects them in a central platform. These two separate channels, shown in Figure 2, for gathering internal and external data from industrial robots are discussed further in the chapter. It is also shown in Figure 3, which shows the layout of data flow in a simplified Open Systems Interconnection (referred to as OSI) schematic diagram. The system can also work in reverse, i.e., sending data from the cloud platform to the embedded device and robot controller. The entire solution was designed for the robotic cell to gather information about industrial robots and their environment. The chapter further discusses the chosen system Thingworx for collecting, processing and visualising the gathered data.

Collecting data from robotic cells is critical to understanding the influences that negatively affect their effectiveness. The limited ability to collect data from the robot’s controller leads to the use of multiple external data acquisition devices. Developing a wireless edge device for industrial robots is advantageous in several ways. By removing the wire from the device, or at least minimising its use, the robot’s operation is not disrupted. At the same time, installing these devices on industrial robots and systems similar to them is much faster. So the need to interrupt the robot’s work is only necessary for a few minutes, and the device’s re-deployment takes a matter of moments. Its ease of use also does not require a qualified person to operate the device. The device is also reusable for multiple robotic devices in different environments.

Our system is designed for a workplace consisting of one industrial robot. However, it could easily be replicated to multiple devices or an entire production line. This would allow the operator to have an overview of the entire production process and, thus, control it efficiently. Furthermore, the system could learn to autonomously schedule downtime in the plant or limit production more efficiently. Control of the production process could also be improved.

### 4.1. Internal Data Flow Settings

This communication channel is set for gathering internal data from the ABB IRB robot controller IRC5. The controller runs the industrial robot and its tasks, and it is also responsible for the robot’s safety. Creating a working task for the robot is done using RobotStudio software, which is an original software from the developer of ABB robots. Besides the development of the actual work task that the robot will perform, you can create another task, which will run independently of the task performed. This was used to create a task, which is cyclically reading information about the robot’s joint parameters and writing them down. It is critical to use different tasks for performing this procedure to maintain the continuous flow of the gathered data. By default, it is possible to read the robot joint’s speed, position and torque in real-time. This information is also that which we are sending to the cloud platform. After the robot controller reads the data, they are converted to the data type PERS, which makes them visible on the network. So while the robot is working, the data collected from the robot’s joints are published on the OPC UA server. This method is used because of the simplicity of the implementation in the industrial sphere [50]. In the same way, all the IRC5 that are in the laboratory are connected, which means that in future, they can be easily connected to the proposed system.

After data are published through the Open Platform Communications United Architecture (referred to as OPC UA) server on the network, they are sent to industrial middleware PTC Kepware. Middleware in IoT is usually used to ease the development process by integrating heterogeneous computing and communications devices and supporting interoperability within diverse applications and services [51]. In our case, the main task of this software is to ensure data readability for the end platform. The PTC Kepware middleware was specially chosen for this application because of the compatibility between chosen cloud platforms. The significant advance in this solution is that we can also send data back to the robot´s controller. This is achieved by creating a communication channel, which Kepware provides. Figure 4 shows the data flow for the internal data.

### 4.2. An Embedded Device for External Data Flow

While the internal solution sends information about the robot’s joint parameters, this solution gathers data on the robot itself, which is called the edge of the Internet of Things [52], for this purpose, and an embedded device was developed, which is described below. The reason for developing this device was to obtain more information about the industrial robot and its working environment. The main requirement was to place this device on the robot. This allows us to be as close as possible to the parameters under investigation and is also one of the main reasons to use embedded devices [53].

The core of the entire embedded device is a microcontroller. The choice of the microcontroller itself was made for the simplicity of the application and the ease of use. Also, compared to microcomputers, microcontrollers are smaller and, thus, more suitable for deployment on the Internet of Things edge. Before the final microcontroller was selected, testing of a few microcontrollers was carried out, which revealed several parameters that are crucial for these devices and their use on an industrial robot. We observed parameters such as the Wi-Fi and Bluetooth range, the microcontroller’s processor, dimensions and so on. The selection of a microcontroller was also subject to the requirements for appropriate software support in code development, development environment support, and library functions. It is also based on the possibility of using available microcontrollers at the workplace, our own development experience, the installed IDE environment, and the support of individual communication protocols. After evaluating the requirements and most of the development experience, we decided to implement the listed microcontrollers in Table 4 and also in Figure 5. The computational performance of the device was also monitored due to edge computing, which is used as a small extension of the final device. This parameter was investigated because of the smoothness of the operation of collecting and sending data to the cloud. It is also important for the future development of the device, where a higher computational load on the microcontroller is expected. Besides this, there are the dimensions of the microcontroller, which in later design phases, determined the size of the whole device.

After testing all these microcontrollers, the Nicla Vision was chosen as the core for the embedded system, see Figure 6. Besides the main requirements, this microcontroller is also equipped with many sensors, such as an accelerometer and gyroscope, which are used to measure these parameters directly on the robot. The advantage is that the device’s overall dimensions are small due to the size of the microcontroller. The device’s small dimensions were also a requirement for the entire system. This fact makes it very easy to place the device anywhere on the robot’s cover without compromising the safety of the robot’s work performance. Previously tested devices were also on the robot but had to be connected to a power supply; and because of that, they had to be placed in one location. Due to this and their large size, some operations have resulted in collisions with the robot.

The proposed device is not connected to the power supply but has its own battery. The Li-Pol 3.7 V battery was chosen for the device. The battery has a capacity of 560 mAh. Using its own power supply gives the device more flexibility compared to previous devices. A theoretical calculation of battery life, while the program was running was performed. A program was run on the device to read the x, y, and z axes acceleration values. The values are read from the sensor in a loop, written and published using MQTT. Therefore, the Wi-Fi module must also be enabled on the device. In this operation, the total average consumption of 120 mA was measured. With the selected battery capacity, the device can operate in this mode for 4.67 h.

When designing the battery for the device, the main requirement was minimalistic dimensions. For this reason, a battery with a smaller capacity was chosen. However, depending on the nature of the measurement, a battery with a larger capacity could be selected, and the device packaging in which it would be placed could be adapted accordingly. For example, if we chose a 12,000 mAh battery, the total working time of the embedded device would increase to 100 h, i.e., longer than four days of operation.

The entire system is then packed into a designed case made using 3D printing. The overall dimensions of the proposed device are only 52 mm long, 42 mm wide and 12 mm high. The system can be placed all over the package in the image of the whole industrial robot. The observation of the parameters, then, actually occurs at the source. For most of our measurements, the proposed device was placed on the fifth axis of the robot, see Figure 7. An external antenna is added to the device to improve the Wi-Fi signal range, so there are no more signal dropouts while the robot is working.

### 4.3. Sending Data from an Embedded Device

The internal network is used for both data flows. For the proposed embedded device, we use a dedicated network that has been reserved for IoT devices. This solution is there to prevent an overload in the network. Collected data are sent to the cloud platform using the MQTT protocol, whose function is shown in the diagram in Figure 8. The MQTT is suitable for this solution as it is a lightweight protocol designed for M2M communications in networks similar to those we use in our laboratories. At the same time, this communication protocol does not require as much computing power and, thus, the resulting battery consumption is reduced. The protocol works on the publish/subscribe principal. Basically, the client sends a message with a unique address to the MQTT broker, and the subscriber is able to retrieve this message using the same unique address [54].

After the data are published to the MQTT broker, they are downloaded to the cloud platform. The downloaded data are then stored on the cloud and ready for further processing. This data transfer procedure from the designed device proved efficient and sufficient for our application. The code designed to collect sensor data works in a loop and also publishes data to the broker in a loop. Whenever a new message appears on the server, the cloud platform downloads it.

A measurement was performed in which we examined the amount of data transferred using the MQTT protocol to the cloud platform from embedded devices. The robot’s X-axis, Y-axis and Z-axis acceleration was monitored, measured, recorded and published from an embedded device in a constant loop to the server. The measurement was performed using the Thingworx platform, which counted the amount of data received from the embedded device. The measurement was carried out with the robot switched on and performing the work operation. The recorded waveform was plotted into a graph shown in Figure 9. A time period of one minute was measured. From the chart, it can be seen that the total data volume for one minute is 3780 bytes for each variable. Thus, the total amount of data transferred is 11,340 bytes. The size of one piece of information is 10 bytes; from this, we can calculate the sampling rate, which is 6.3 samples per sec. The graph also shows that the data are received consistently and that there are no significant fluctuations.

Upon closer examination of the data, it was found that there is little data loss when transferring from the embedded device to the cloud. The total loss was 3.3% of the total volume per hour. This phenomenon is primarily due to network overload. This is due to a lab network where multiple users may connect during the day. In our case, where the robot performs cyclic work, i.e., it repeats the same task over and over, the data loss is a relatively minor issue, and the error rate can be filtered out by measuring multiple cycles. In the long term, we are preparing the system for the transition to the 5G network, which guarantees lossless data transmission to some extent.

### 4.4. Reversed Data Sending

The same communication paths described in the previous subsections are used for sending data from the cloud to the embedded device and robot’s controller, see Figure 10. In the case of the proposed embedded system, it wirelessly sends instructions to it about what program to run and what parameters to use for measurement. The system in the embedded device works by switching on, connecting to the network and MQTT broker, switching to listening mode and waiting for an instruction sent from the Thingworx platform. The entire device is thus set up for remote control; so, there is no need to interfere with the robot’s operation or stop it when a measurement change is necessary.

The communication channel between the cloud platform and the robot’s controller is also used for sending instructions. As the cloud platform process collected data, it is eventually able to send instructions to the robot’s controller. This is also one of the most significant benefits of the whole system, where the robot’s behaviour can be modified in real-time via a cloud platform based on data collected from various sources.

## 5. Processing Data

The data collected from all sources are stored on a server where space is reserved for them. They are accessible on the Thingworx cloud platform, where an interface was created to display the measured data, so the user can immediately view the measured data and their dependence on each other. The data display is shown in Figure 11. Then, it is possible to export each measured parameter and save data to the .csv file. This is also possible in reverse to import a .csv file with data from an external source to compare them with our measured data. Besides this, it is possible actually to work with data in Thingworx. This is currently used for minor autonomous data handling, where temperature information is averaged, and alerts are automatically sent to the user based on the system’s observation and evaluation. For example, this functionality was also used to measure the total volume of data received on the platform, as demonstrated in Figure 9. In exactly the same way, robot behaviour will be controlled in the future. As with temperatures, the system will automatically assess the situation and send the information to the robot’s control unit.

Besides the fact that the data are processed in the Thingworx platform, little processing occurs on the embedded device. Primarily, it is about editing and sorting data to reduce the requirements for sending them. For example, the “Running average” function averages the collected data and, thus, reduces the amount of data required for transmission.

## 6. Conclusions and Future Work

In this article, we describe the current state of work on a large-scale project, the goal of which will be the autonomous work of robots. Specifically, the paper deals with a proposal for implementing an embedded system for data collection and evaluation, which is attached directly to the robot’s arm. Work on the development of the entire system is ongoing. The project aims to bring the system to an autonomous state, in which it will evaluate the parameters that affect the industrial robot and then adjust the robot’s behavior based on them. This would increase the efficiency of the robot and the quality of the work performed. In the example of our workplace, which is designed for the operation of deburring workpieces using a robotic arm, the quality of the final workpiece depends on the oscillation of the robot. By compensating based on data acquisition, using our system, the docking would be reduced, thus increasing the quality of the result and eventually the efficiency of the whole workplace. At the same time, work will be carried out on the system to monitor and analyse the collected data in the long term. It should therefore examine patterns, deviations, and anomalies in data and warn, for example, of an impending failure. Thanks to this, the failure rate of the robotic system would be prevented, which would lead to a reduced need for maintenance and overall to the elimination of negative situations associated with the degradation of the robot. We have to ensure the collection of information and various measured values by implementing an additional embedded system, which of course, is not part of the robot. We are creating this embedded system to collect other parameters, such as the influence of the external environment of the workplace. The developed system is miniature, with its own power supply, compatible with wireless technologies, reusable for all parts of the workplace, and with the possibility of expansion by other types of measured physical quantities using additional sensors.

We describe the current state of the discussed issue from the point of view of available studies, a description of the IoT technologies used, and their placement in the general proposed IoRT framework in the article’s introduction. Then, we presented the robotic workplace in more detail, including the hardware and software used. As the work described in this article is part of a significant technological unit, the types of robots, controllers and ABB Robot studio software used and PTC Thingworx are fixed, and we do not explore other technology design possibilities. The core of the entire system is the ABB IRB 1600 industrial robot workplace. This system must be regularly checked and maintained to avoid undesirable situations. We have described several software environments and server tools necessary for programming the robot’s work operations and connecting the robotic systems of the entire workplace. We have also described the communication channels and the operation of the robot’s control unit.

We designed a comprehensive solution for collecting information about industrial robots and their environment for the entire robotic cell at the workplace. A significant advance of this solution is that we can also send data back to the robot’s control unit. This is achieved by creating a communication channel provided to us by the chosen Middleware. This communication channel between the cloud platform and the robot controller is also used to send instructions after an overall analysis of the collected data. This is also one of the most significant benefits of the whole system. The robot’s behavior can be adjusted in real time through the cloud platform based on the data collected from various sources. When selecting a microcontroller, we considered the circuit structure, communication capabilities, computing power, strength and quality of wireless transmission given by testing in real operation, dimensions, consumption and power supply options. In addition, the device’s small size was also a requirement for the embedded system. This fact makes it very easy to place the device anywhere on the robot’s enclosure without compromising the safety of the robot’s work performance. The resulting implemented system runs on a Nicla microcontroller. Still, in the future, we can replace the microcontroller according to the current trend in the field of microcontroller development, which we are actively following. During the implementation of the embedded system, a number of measurements were performed in live operation, the results of which, including the amount of data obtained, were mentioned. These obtained measured values, which are in the range of six measured values per second, are fully sufficient for our workplace. However, work on the entire ecosystem will continue. For the core part of the subsequent work, the project will continue with the implementation of the 5G network, which will enable a great leap in speed and a significant reduction in response time on the network elements of the architectural model.

## Figures and Tables

**Figure 1 micromachines-14-00113-f001:**
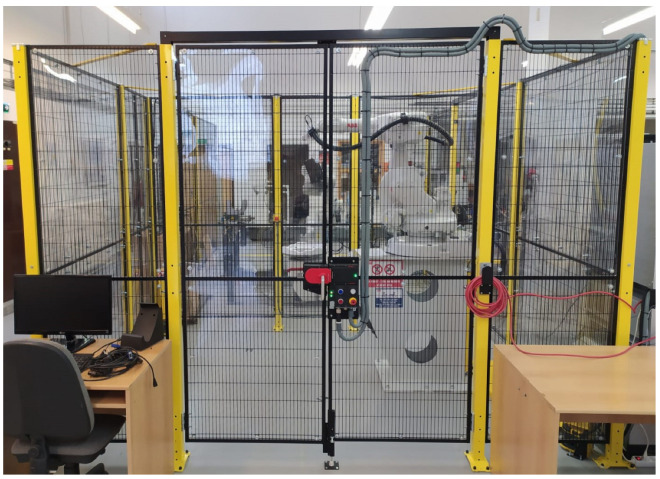
The picture shows a robot cell with an ABB IRB1600 robot.

**Figure 2 micromachines-14-00113-f002:**
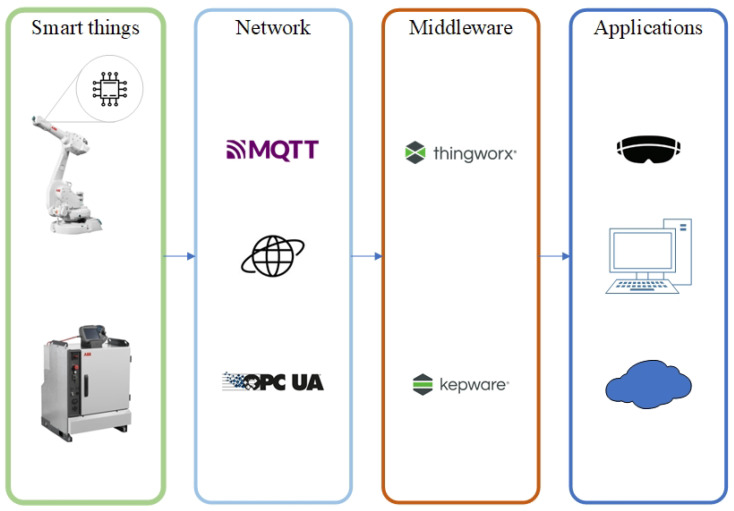
The frame of our proposed IoT system.

**Figure 3 micromachines-14-00113-f003:**
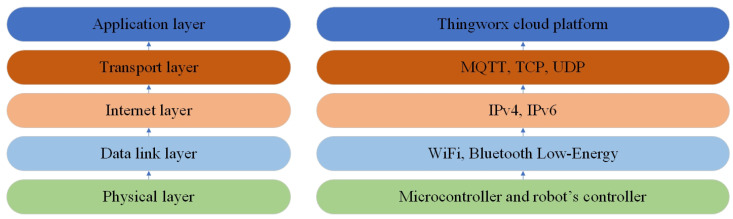
This figure shows the layout of the system in a simplified OSI diagram.

**Figure 4 micromachines-14-00113-f004:**
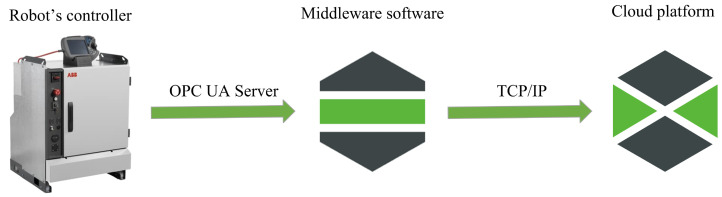
This figure shows how the data contains information about robot joint parameters.

**Figure 5 micromachines-14-00113-f005:**
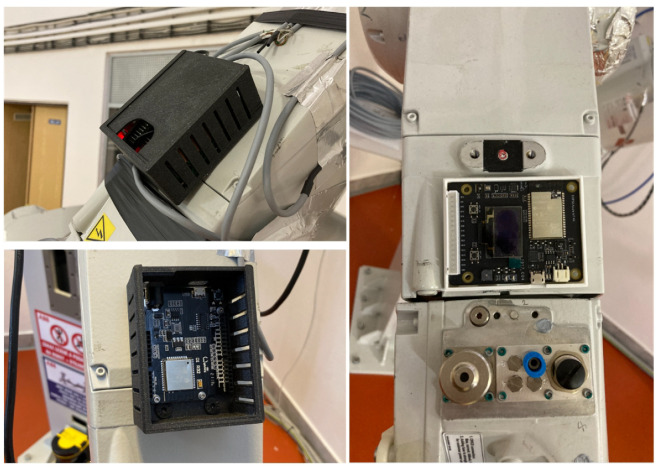
The figure shows the tested solution with ESP8266 (**left**) placed on the robot. This solution proved to be inadequate for our application. The dimensions of the whole device were quite large and, in some cases, prevented the robot from working safely. There is also the problem with the Wi-Fi signal range, which fluctuated during the robot’s operation. Solution with ESP32 (**right**).

**Figure 6 micromachines-14-00113-f006:**
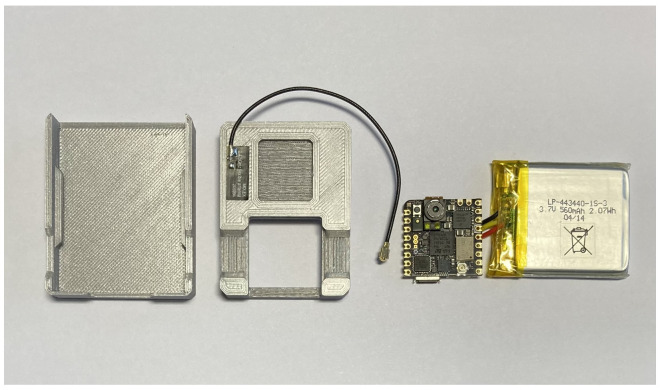
The figure shows the embedded system and its components. The plastic cover holds all the components together and protects them.

**Figure 7 micromachines-14-00113-f007:**
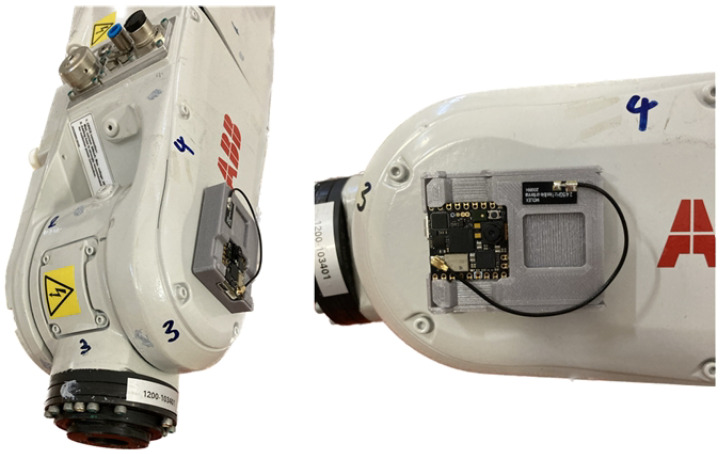
This figure shows the final version of the proposed embedded system mounted on an industrial robot. The device is mounted on the fifth axis of the robot, where it senses the parameters of the robot and its surroundings.

**Figure 8 micromachines-14-00113-f008:**
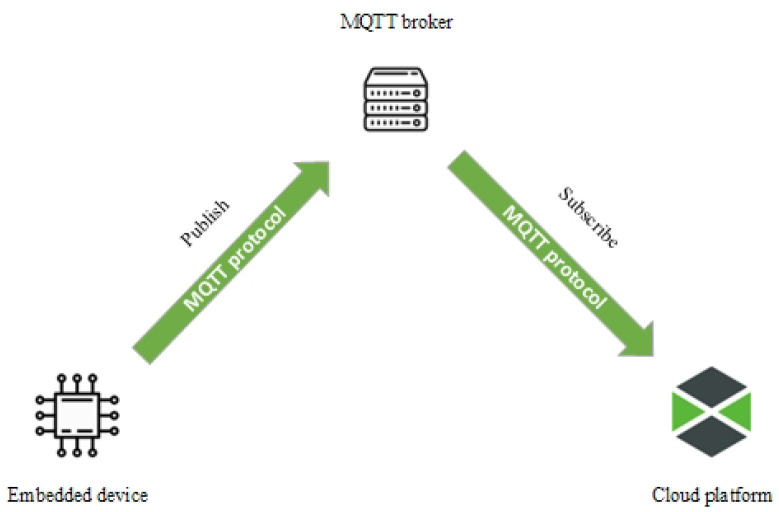
MQTT protocol uses publish/subscribe method to send messages.

**Figure 9 micromachines-14-00113-f009:**
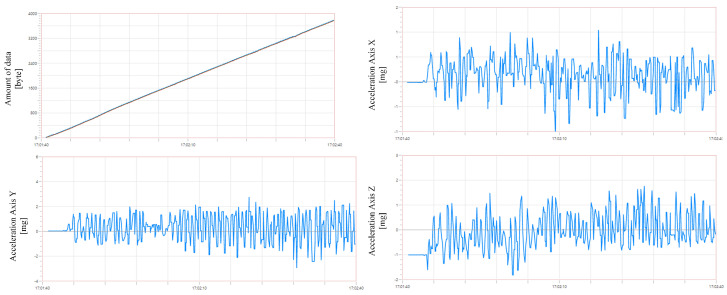
The figure shows a graph (left corner) generated in Thingworx to represent the amount of received data from the embedded system in time. The other graphs show acceleration data in the x, y and z axes.

**Figure 10 micromachines-14-00113-f010:**
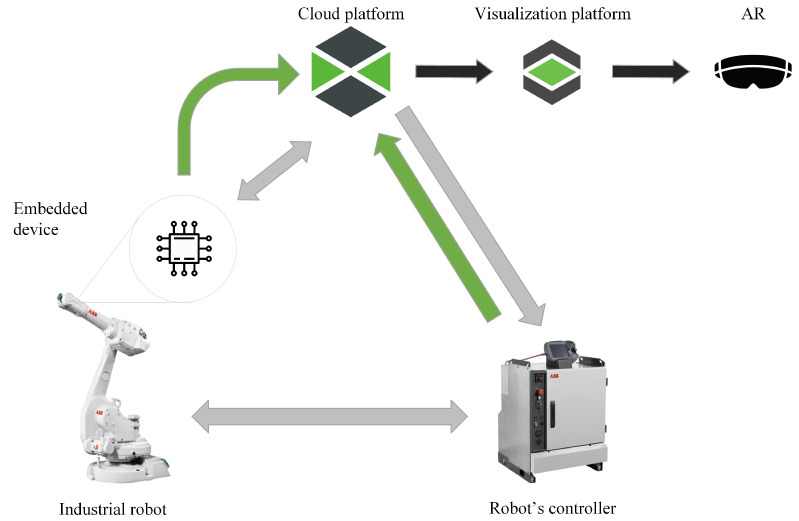
This figure shows how the data flow in the entire system. The data flow from two sources, where one of them uses the embedded device, which was developed for this purpose. The diagram shows the paths for the data flow containing the monitored parameters (green arrows) and paths for the data containing the control information (grey arrows). The black arrows, then, show the path for the data that are visualised.

**Figure 11 micromachines-14-00113-f011:**
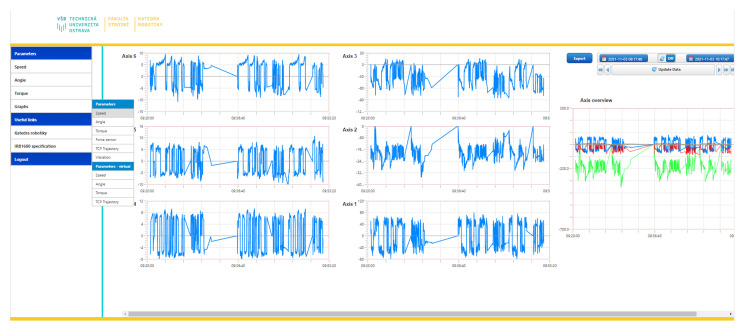
The figure shows that data can be displayed in a web browser. This is used to quickly view the measurement progress or to check the robot parameters. A great advantage is the ability to preview this from anywhere.

**Table 1 micromachines-14-00113-t001:** This table contains an overview of articles dealing with the Internet of Robotic Things.

	Outline of the Works	General Overview	Architectural Layer Model	Communication Protocols	Approaches/Chalenges/Future Trends	Application
[12]	IoRT technologies and application study	✓	✓		✓	✓
[13]	Approaches and challenges in IoRT—review	✓	✓		✓	✓
[14]	IoRT architecture, protocols, cloud robotic platforms	✓	✓	✓	✓	✓
[15]	Smart technologies in Industry 4.0, smart domains, IoRT components	✓	✓		✓	✓
[16]	IoRT concept, emerging technologies, architecture, application and robotic roadmap	✓	✓	✓		✓
[28]	An analysis of the Industrial Internet of Things (IIoT)	✓	✓	✓		✓
[29]	A systematic literature review of IIoT	✓	✓	✓	✓	
[30]	IoT, IIoT comparison, Industry 4.0, architecture, connectivity and standardization of IIoT	✓	✓	✓	✓	
[31]	The study of IoRT concept, issues and challenges	✓	✓		✓	✓
[32]	Architecture of software-defined IIoT in the context of Industry 4.0	✓	✓			✓

**Table 2 micromachines-14-00113-t002:** This table contains an overview of articles dealing with microcontrollers and the development platforms.

	Outline of the Works	Platform Comparison	HW Structure Analyze	Software Tools and Environment	IoT Practical Implementation
[33]	Comparison of Arduino IDE compatible platforms for digital control	✓	✓	✓	
[34]	ESP32 microcontroller module comparative analysis	✓	✓		✓
[35]	STM devices and platforms for IoT		✓	✓	
[36]	Analytical comparison of using low-cost microcontroller modules in embedded systems design	✓	✓		✓
[37]	Source code performance investigation of MCU programming language		✓	✓	✓
[38]	Microcontrollers for IoT—architecture, research topics and future trends	✓	✓	✓	
[39]	Comparison of microcontrollers and practical IoT implementation	✓	✓		✓
[40]	A review of microcontroller unit for wireless sensor node platforms	✓	✓		
[41]	Practical implementation of ESP32 microcontroller in IoT application	✓	✓	✓	✓
[42]	Control of robotic arm moving via IoT with ESP32		✓		✓

**Table 3 micromachines-14-00113-t003:** This table contains an overview of articles related to concepts used in our work.

	Outline of the Works	Cloud Solution	Thingworx Using	MQTT Protocol	System Design & Development	Robotic Applications
[1]	IoT common standards, technologies, and protocols—survey	✓		✓		
[19]	Comparison of IoT middleware platforms; Similarities and differences	✓	✓		✓	
[20]	IoRT data exchange techniques detailed study	✓		✓		✓
[43]	The system architecture of cloud robotics; Development of cloud robotics	✓			✓	✓
[44]	Cloud-based system research; Cyber-physical systems	✓		✓	✓	✓
[45]	IoT platforms comparison, functional requirements, comparative techniques	✓	✓	✓		
[46]	Current IoT protocols, evaluation, and comparison			✓	✓	
[47]	IoT protocols, detailed description of MQTT use, MQTT brokers	✓		✓		
[48]	Using MCUs to control a robotic arm via the cloud	✓			✓	✓
[49]	practical use of MCU and IoT system for monitoring	✓			✓	

**Table 4 micromachines-14-00113-t004:** This table shows the monitored parameters of microcontrollers used as a basis for an embedded device on an industrial robot.

Parameters	ESP8266	ESP32	Nicla Vision
Processor Type	L106	Xtensa LX6	STM32H747AII6
Processor	160 MHz	240 MHz	480 MHz
Wi-Fi	802.11 b/g/n	802.11 b/g/n	802.11 b/g/n
Wi-Fi Speed (Up to) [Mbps]	72.2	150	65
Bluetooth	no	Bluetooth 4.2	Bluetooth 4.2
Operating Current (Average value) [mA]	80	180	100
Integrated Battery Charger	no	no	yes
Dimensions of MCU (Length, Width) [mm]	46, 26	68.58, 53.34	22.86, 22.86
Number of I/O Pins	12	12	12
Integrated Sensors	no	(ESP32-Azure IoT kit) motion sensor, magnetometer, barometer, light sensor, temperature and humidity sensor	accelerometer, gyroscope, camera, microphone

## Data Availability

The data that support the findings of this article are available from the corresponding author upon reasonable request.

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
