# Peer review of "Implementation of an Embedded System into the Internet of Robotic Things"

_micromachines, 2022, doi:10.3390/mi14010113_

Round 1

Reviewer 1 Report

System is under devlopment.  Its not fully autonomous right now.  Its more the DAQ for robot rather than any correction and learning from the captured data. you can use AI/ML/DL for data processing. No estimation or predication? 

Line 13: Write full form of IoRT. (First time only) 

Line 21: IoT full form? go through the complete article. New abbrivation should be define when it appear first time in article. 

Figure 1 text citation is missing

Figure 3, 4,5,56,7,8,10,11 text citation is missing

Line 214: Figure 7 or Figure 9? 

line 246: Figure text citation iswrong

Figure 11: Snap shot is not proper. Please change.

Figure 9 and 11: Not clear. Axis information is not visible.

Line 269: " This would increase efficiency of the robot"?  how much? how you evalute the efficiency of any task? is it used for multitasking porpose? 

Line 273: How much is failure rate? What is the data set size. How to reduce the redandant data? 

Line 275: What are the diffrent parameters are used to calculate the degradation of robot?

Line 279: Power supply specification? 

Conclusion part is dicussing the part related to introduction. No statictical or numerical or mathamatical analysis is available in paper or firm conlcusion or outcomes. Its not review paer. Its a research paper. so you have to use more ref. of implementaion/research rather than review papers. 

Author Response

The answers for the reviewer are attached in a pdf file.

Reviewer 2 Report

The paper is well written in term of technical overview. I just have some comments that might improve the overlook of the paper.

1.       The abstract provides multiple ideas, once it mentions it just a description of the use of embedded systems in the Industrial Internet of Things and its benefits for industrial robots, then it says it’s a case study, finally, its mention that proposed system. I suggest the authors to be precise in the abstract about what they are doing.

2.       In the introduction, line 17 mention that it is a chapter please correct to a proper name such as “section”.

3.       The authors are suggested to rewrite the introduction, the literature review is being over the investigated problem to draw the problem statement of what is the research approaching.

4.       In figure 4 please use high resolution image and mention the copy right of the image if it not yours.

5.       The authors are suggested to compare multiple communications/microcontroller systems.

6.       The authors are suggested to expand the system to hold two robotic and how that would improve their decision-making.

7.       The authors are suggested to re-organize the paper to hold main idea, evidence, analysis etc.

8.       The figures need to be high quality and zoomed in.

Thank you.

Author Response

(The authors gave the same response as above.)

Round 2

Reviewer 1 Report

Fig 4: Missing

Fig 11: Missing

Only 32 refrenced used?  Justify.  Base of lirature is missing. Add comparitive analysis of controllers and other software and hardware used in proposed system. 

Need to write seprate Litrature review section: 

Author Response

The reply to the review report is in the attached doc file.

Reviewer 2 Report

The comments need be addressed carefully.

Author Response

(The authors gave the same response as above.)

Round 3

Reviewer 2 Report

Thanks for the authors for making the efforts of responding to our comments.